# Sorption and Photocatalytic Characteristics of Composites Based on Cu–Fe Oxides

Alexander Agafonov [1], Anastasia Evdokimova [1], Andrey Larionov [2], Nikolay Sirotkin [1], Valerii Titov [1] and Anna Khlyustova [1,*]

1   G. A. Krestov Institute of Solution Chemistry of RAS, Academicheskaja Str., 1, 153045 Ivanovo, Russia
2   Department of Inorganic Chemistry, Ivanovo State University of Chemistry and Technology, Sheremetevsky pr., 7, 153000 Ivanovo, Russia
*   Correspondence: kav@isc-ras.ru

**Abstract:** Plasma ignition in the volume of liquid solution/water initiates the chemical activation of the liquid phase (formation of chemically active particles) and the sputtering of electrode materials, which leads to the formation of nanostructured materials. In this work, the synthesis of structures was carried out by means of underwater plasma excited in water between electrodes composed of different materials. The polarity of the Fe and Cu electrodes was varied at two plasma currents of 0.25 and 0.8 A. The kinetics of the sorption and photocatalysis of three dyes (Rhodamine B, Reactive Red 6C, and Methylene Blue) were studied. According to the results obtained, the polarity of the electrode material has a greater effect on the phase composition than the plasma current. The sorption process can be limiting depending on the type of dye and phase composition. The sorption kinetics can be described by various models at different stages of the process. Photocatalytic studies have shown that the complete decomposition of the three dyes can be achieved in 15–30 min of irradiation.

**Keywords:** photocatalytic activity; iron; copper; plasma; combustion





## 1. Introduction

To date, sorption methods for removing organic and inorganic contaminants from wastewater from industrial enterprises are the cheapest [1]. Despite this, new, promising composites have recently been developed that can be used as multifunctional materials, for example, sorbents and catalysts. Among such materials, materials with high sorption capacity and photocatalytic properties are of great interest. This simultaneously allows both removing the pollutants and decomposing them into harmless compounds.

There is a wide variety of methods for obtaining photocatalysts (physical, chemical, biological) [2,3]. The best-known photocatalyst is titanium dioxide, which works excellently by irradiating UV light. Its efficiency is reduced when irradiated with visible light. To increase its effectiveness, titanium dioxide is doped [4]. The best-known photocatalyst is titanium dioxide. However, for it to work effectively in visible light, it is doped [4]. In addition, it has low porosity. Iron oxides have a high porosity [5–7]. In several studies, it was found that iron oxides are considered as the photocatalysts for the various classes of dye removal [8–11]. Copper oxides are regarded as potential photocatalysts due to the narrow band gap [12]. It is possible to increase sorption or photocatalytic characteristics using binary oxide or ternary systems [13–17].

Synthesis methods can play a main role in the formation of the surface properties and structure of the obtained materials. Cu–Fe binary oxides are synthesized by co-precipitation [18], hydrothermal [19], sol-gel [20], and various variations of the combustion method [21–23]. The chemical solution combustion method makes it possible to obtain structures with the desired shape and with a large specific surface area [24,25]. Previous studies have shown that the underwater plasma method allows for obtaining pure oxides, doped oxides, and binary systems [26–28]. In this paper, we compare the characteristics of

materials obtained by chemical combustion and underwater plasma and also study their sorption and photocatalytic properties. The novelty of the presented work is the use of less refractory metals as electrodes in the plasma system for creating binary oxide systems and also studying the possibility of using the obtained materials as sorbents.

## 2. Materials and Methods

### 2.1. Methods of Synthesis

The synthesis of oxide was carried out in the glass cell using underwater plasma (Figure 1a) [29]. Copper and iron rods (99.9 % purity, Cvetmetsplav, PJSC, Russia) with a diameter of 0.9 mm were used as electrodes in the experiments. These rods were placed in a ceramic tube. The non-isolating length of the electrodes was 3 mm. The electrodes were immersed in distilled water. Direct current (DC) underwater plasma was ignited between electrodes in water using the homemade DC power supply with an output voltage of up to 5 kV and a 5 kOhm ballast resistor. The experiments were carried out at two plasma currents. The polarity of the electrode materials was changed. The time of treatment was 5 min. Formed suspensions were centrifuged, and precipitates were dried at 100 °C for 3 h.

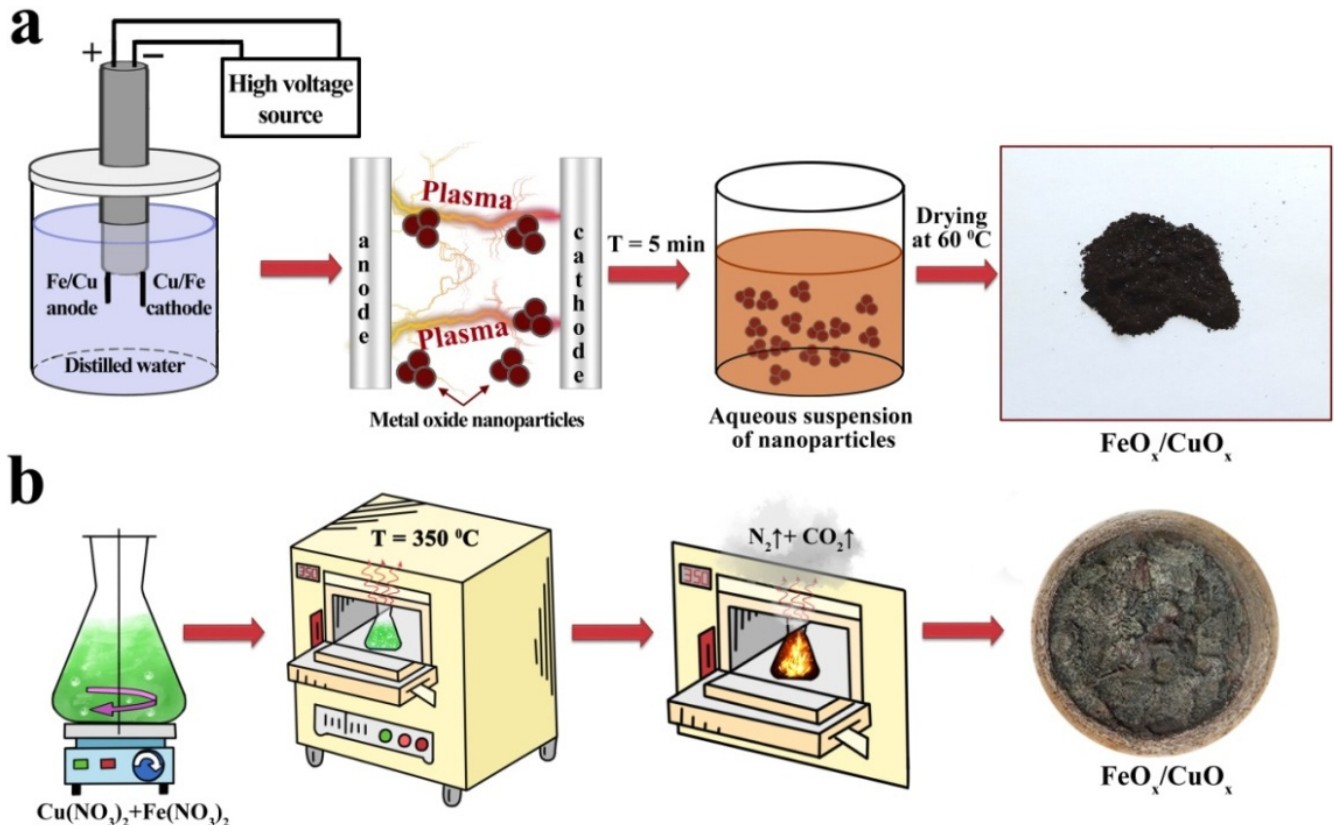

**Figure 1.** Scheme for the synthesis of metal oxides by the underwater plasma (**a**) method and the combustion method (**b**).

Additionally, for comparison, the synthesis of binary oxide structures was carried out by the combustion method (Figure 1b). Copper and iron nitrates (99.9%, Sigma-Aldrich (St. Louis, MO, USA)) were dissolved simultaneously in deionized water. Then, citric acid (99.9%, Sigma-Aldrich (St. Louis, MO, USA)) was added to the solution as fuel. The solution was kept in an oven at 350 °C. Initially, the solution boils and loses water. Next, the dehydrated mixture decomposes with the release of nitrogen and carbon dioxide. After reaching the point of the spontaneous combustion of the solution, loss of water, and release of gases, a black powder is formed.

## 2.2. Characterization

The average size of the formed particles and their zeta potential values were measured by the Dynamic Light Scattering method (Malvern Zetasizer Nano, Malvern, UK). The phase composition was studied by X-ray diffraction (XRD, X-ray diffractometer D2 PHASER). A chemical composition and surface morphology were investigated using scanning electron microscopy (SEM) (ThermoFisher, Czech Republic). The thermal analysis of the samples was performed by using the thermomicrobalances TG 209 F1 Iris (Netzsch, Germany) under a continuous nitrogen flow (30 mL/min) and heating from 25 °C to 900 °C at a rate of 10 °C min$^{-1}$.

## 2.3. Photocatalytic Activity

The photocatalytic performance of the synthesized structures was estimated by the rate of destruction of the dyes under UV light irradiation. The detailed procedure was described elsewhere [29]. The water-cooled quartz jacket was immersed in the cylindrical vessel with a dye solution. The source of irradiation was placed in the middle of the quartz jacket. The high-pressure 250 W mercury lamp was used as the source of light irradiation. The light intensity of the mercury lamp in the spectral region of 280–400 nm was 75 W/m$^2$. The effective mixing of a dye solution and powder was provided by a magnetic stirrer. Air was bubbled through a solution to provide a constant concentration of dissolved oxygen. In total, 0.03 g of powder was dispersed in 500 mL of dye-aqueous solution with a concentration of 1.2 mg/L. The mixing suspension was carried out by a magnetic stirrer. The methylene blue (MB) ($\lambda_{max}$ = 667 nm), rhodamine B (RhB) ($\lambda_{max}$ = 554 nm), and reactive red 6C (RR6C) ($\lambda_{max}$ = 544 nm) were used as the model dyes. The choice of dyes for photocatalysis was determined by the fact that we chose representatives of different classes of dyes. Rhodamine B is a fluorone dye that exists in solution as a zwitterion. Active Red 6C is a monoazo dye, which is an anion in solution. Methylene blue is a thiazine dye. It presents in solution as a cation. The concentration of dyes was monitored by UV-vis spectrophotometer (SF-56) by measuring absorbance intensity at $\lambda_{max}$.

## 2.4. Dye Adsorption

The adsorption of dyes on synthesized composites was studied at room temperature in the experimental setup for photocatalytic tests in the dark. The concentration of dyes was determined by measuring absorbance changes at a maximum of dye adsorption. According to work [30], the adsorption capacity at time (mg/g) was calculated by using Equation (1):

$$q_t = \frac{(C_0 - C_t) \cdot V_0}{m_c},$$ (1)

Here, $C_0$ and $C_t$ are the initial and at the time of irradiation concentrations of a dye (mg/L), $V_0$ is the volume of the solution of the dye (L), and $m_c$ is the mass of the catalyst (g).

The amount of equilibrium adsorption capacity (mg/g) was calculated by the next:

$$q_e = \frac{(C_0 - C_e) \cdot V_0}{m_c},$$ (2)

where $C_e$ is the equilibrium concentration of dye after adsorption.

To analyze the adsorption processes of the dyes, we used Lagergren kinetic models for the catalytic chemical reaction:

The pseudo-first-order model:

$$\ln(q_e - q_t) = \ln q_e - k_1 t,$$ (3)

The pseudo-second-order model:

$$\frac{t}{q_t} = \frac{1}{k_2 q_e} + \frac{t}{q_e},$$ (4)

where $q_e$ and $q_t$ are the quantity of adsorbed dyes in the equilibrium state and at time $t$; $t$ is the time of contact; $k_1$ and $k_2$ are the pseudo-first- and pseudo-second-rate constants.

Additionally, the intraparticle diffusion model was described. The linear form equation is the next.

$$q_t = K_{id} \cdot t^{1/2} + C, \qquad (5)$$

Here, $K_{id}$ is the diffusion coefficient, and $C$ is the intraparticle diffusion constant which is directly proportional to the thickness of the boundary layer.

### 3. Results and Discussion

#### 3.1. Evidence of Electrode's Sputtering

The process of electrode sputtering during plasma exposure is confirmed by changes in the electrode masses (Table 1) and plasma emission spectra (Figure 2). The experimental results of changes in the masses of the electrodes during plasma exposure are presented in Table 1. The data showed that the plasma current does not significantly affect the sputtering rate of the iron electrode at any polarity, while for a copper cathode, the current growth leads to a 20-fold increase in the change in the mass of the electrode. In this case, there is a significant deviation from Faraday's classical law of electrolysis. The small difference in the sputtering rates of the iron anode and cathode can be explained by the passivation of the surface of the iron anode (the formation of a thin oxide film on the surface under the action of hydroxyl ions from the solution).

**Table 1.** Mass change of electrodes.

| Sample | Cathode | Anode |
|---|---|---|
| Fe–Cu 0.25 A | 0.0015 (Fe) | 0.0043 (Cu) |
| Fe–Cu 0.8 A | 0.0019 (Fe) | 0.0113 (Cu) |
| Cu–Fe 0.25 A | 0.001 (Cu) | 0.0019 (Fe) |
| Cu–Fe 0.8 A | 0.0203 (Cu) | 0.0033 (Fe) |

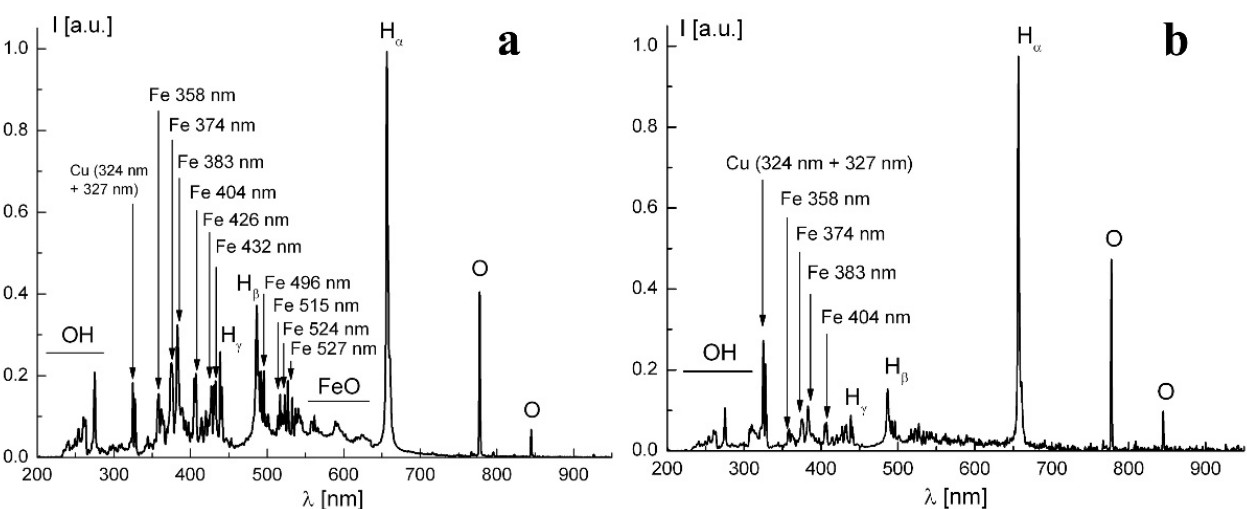

**Figure 2.** Emission spectra of the underwater plasma with Fe anode and Cu cathode (**a**) and Cu anode and Fe cathode (**b**).

This section may be divided into subheadings. It should provide a concise and precise description of the experimental results, their interpretation, as well as the experimental conclusions that can be drawn.

Figure 2 shows the emission spectra of the underwater plasma. The plasma current does not affect the qualitative picture of the spectrum. However, the polarity of the iron electrode affects the shape of the emission spectrum. In experiments with an Fe anode, the spectrum contains more lines of atomic iron, including bands of metastable FeO [31].

Copper is represented in the spectrum only by resonance lines and does not depend on polarity.

### 3.2. Characterization of Obtained Materials

An analysis of particle sizes in liquid suspension showed that an increase in the plasma current leads to the formation of larger particles (Table 2). The larger size of the hydrodynamic radius for particles obtained by chemical combustion (S5) can be explained by the presence of citric acid (fuel/stabilizer) affecting the viscosity of the solution.

**Table 2.** Data of DLS analysis.

| Sample | $D_{av}$, nm | $\zeta$, mV |
|---|---|---|
| Fe–Cu 0.25 A (S1) | 114 ± 12 | 31.0 |
| Fe–Cu 0.8 A (S2) | 169 ± 17 | 15.6 |
| Cu–Fe 0.25 A (S3) | 151 ± 15 | −19.5 |
| Cu–Fe 0.8 A (S4) | 251 ± 24 | 29.5 |
| Fe/Cu oxides (S5) | 317 ± 17 | 17.4 |

Figure 3 presents the X-ray patterns of the synthesized powders. In experiments with underwater plasma with a copper anode, the plasma current does not affect the phase composition (samples S3 and S4). In the case of an iron anode, at a low plasma current, conditions are created for the formation of layered structures of Cu–Fe layered double hydroxides with an admixture of copper oxide (sample S1). As the plasma current increases, a mixture of iron and copper oxides is formed ($Cu_2O_3 + \alpha\text{-}Fe_2O_3 + FeO$) (sample S2). The powder obtained by chemical combustion (sample S5) contained $\alpha\text{-}Fe_2O_3 + Fe_3O_4$ and copper oxides ($Cu_2O_3 + Cu_2O + CuO$). The presence of copper ions with different oxidation states can be caused by the use of only one fuel in chemical combustion [25].

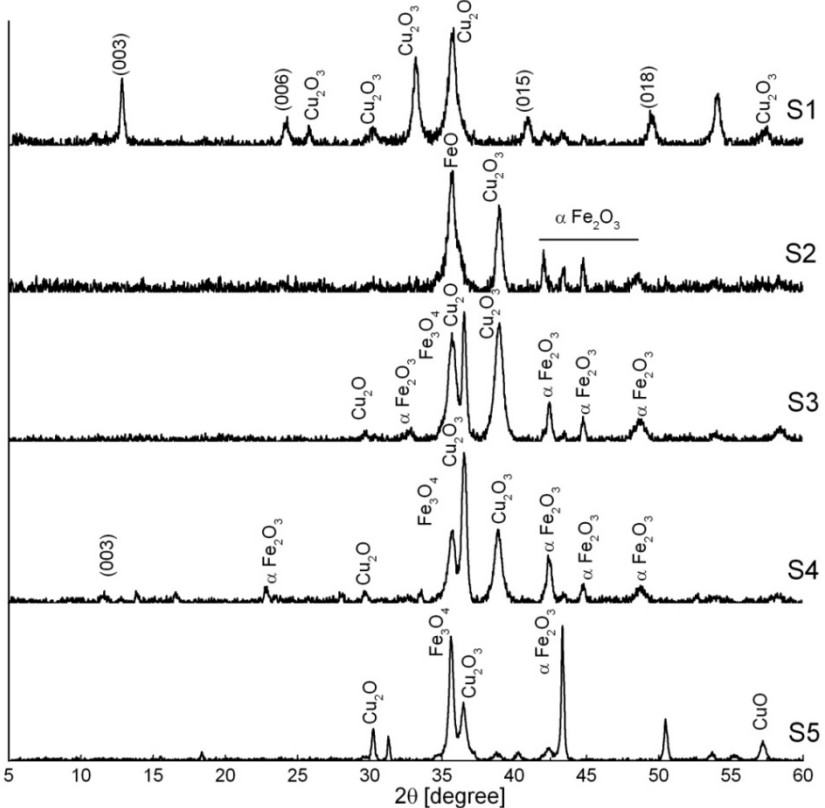

**Figure 3.** XRD patterns of synthesized powders.

Particle size was estimated using the ImageJ software. SEM images of the synthesized powders are presented in Figure 4. The particles are generally round in shape. The exceptions are samples S3, where rhombic-shaped particles predominate, and S4, in which particles of both round and rhombic shapes are present. The estimates of particle sizes are presented in Table 3. As can be seen, the largest particles are present in samples S4 and S5. It is known that the dynamic light scattering method shows the hydrodynamic particle size. In this case, the particle size also includes the hydration shell around it. Thus, the particle size values obtained from the DLS method can be significantly larger than the particle sizes obtained from the results of SEM.

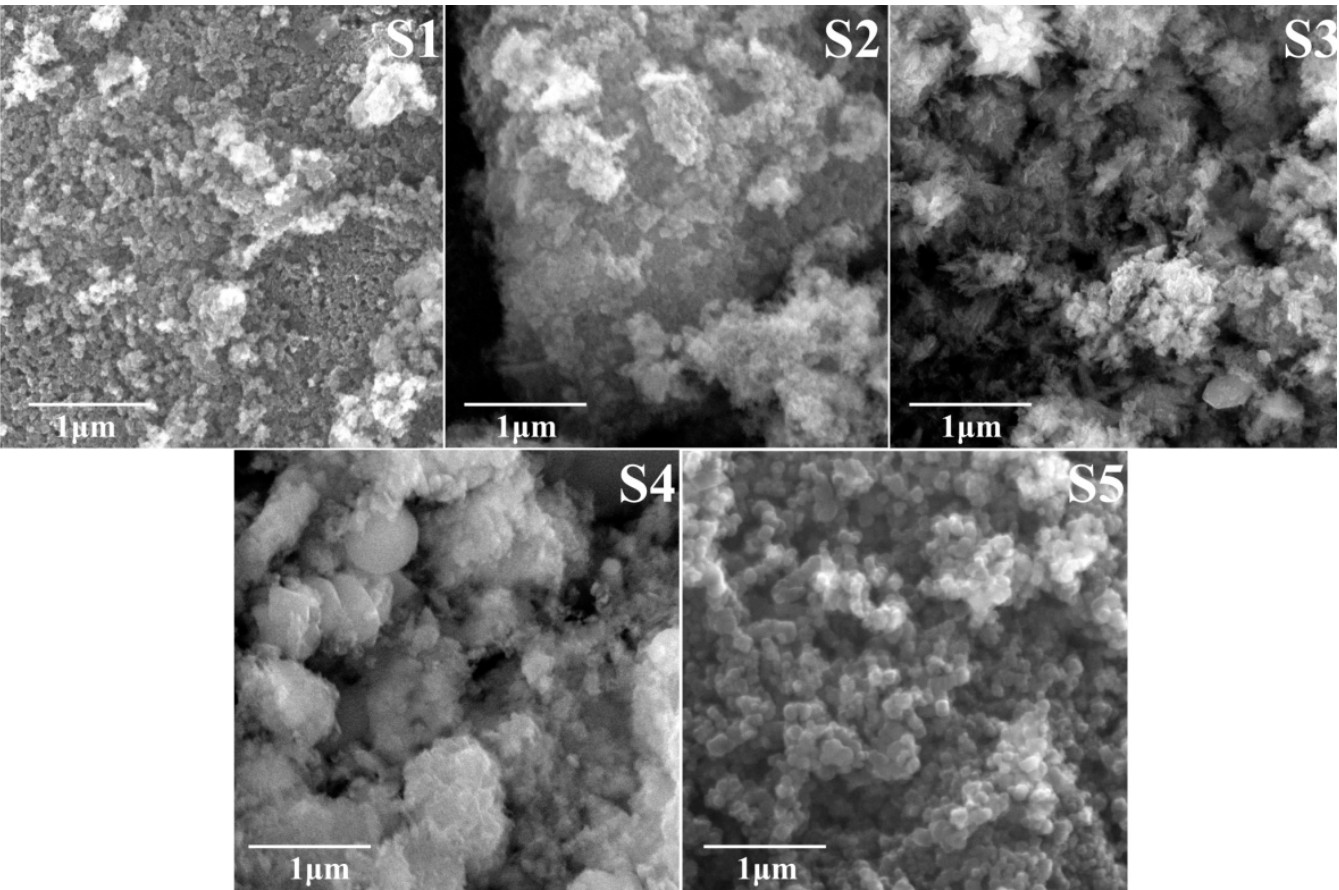

**Figure 4.** SEM images of synthesized powders.

**Table 3.** The size of particles (nm).

| S1 | S2 | S3 | S4 | S5 |
|---|---|---|---|---|
| 50–70 | 65–90 | 110–140 | 130–180 | 120–150 |

Thermogravimetric curves are plotted in Figure 5. It can be seen that sample S2 loses the most mass (25.59%), and sample S5 loses the least (4.63%). For samples S2 and S5, two stages of weight loss were recorded. In this case, for S5, the stages were observed at higher temperatures. For this sample, there are no stages of weight loss due to the removal of sorbed water. This is due primarily to the method of synthesis. For samples S1, S2, S3, and S4, there are stages of removal of physically sorbed water (25–100 °C) and chemically sorbed water (removal of interlayer water for S1) (200–300 °C). The stage at 300–400 °C corresponds to the decomposition of $Cu_2O_3$ and $Fe_3O_4$. Small changes in the mass of the samples in the range of 400–900 °C may indicate the formation of copper ferrite $CuFe_2O_4$ [32].

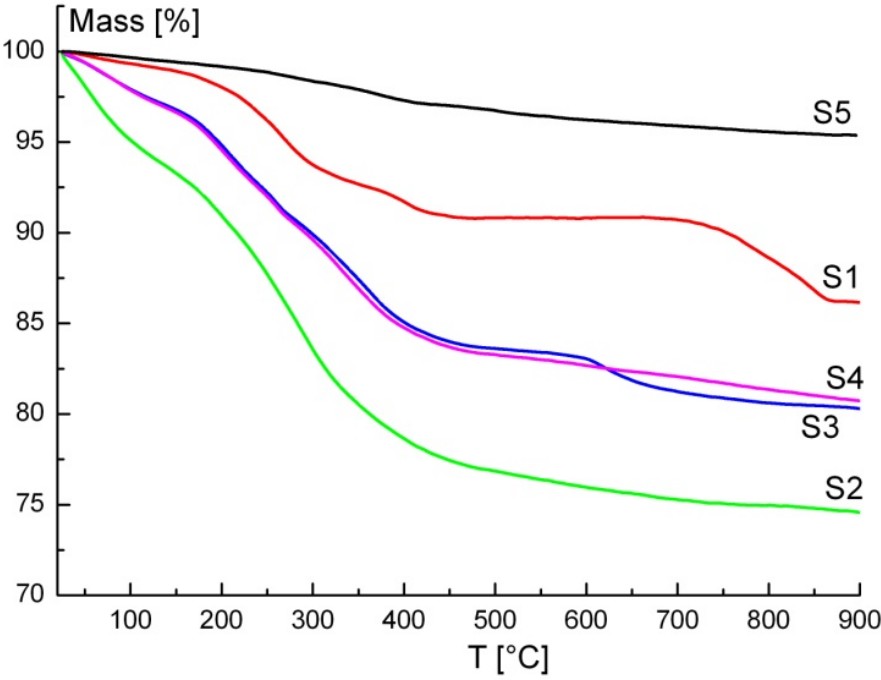

**Figure 5.** TG curves of obtained samples.

### 3.3. Kinetics of Dye's Absorption

Studies of photocatalysis were carried out with the dark phase to determine the role of the sorption stage in the process of photocatalysis. The kinetic curves for the three dyes are shown in Figure 6. The sorption kinetics of the three dyes on the synthesized samples (dark phase of photocatalysis) are shown in Figure 7. The calculated values of the kinetic parameters are presented in Tables 4–6. As can be seen, the S1 sample has the highest sorption capacity with respect to the dye Rhodamine B. The active red dye is more sorbed on samples S2, S3, and S5. In the case of methylene blue, sample S4 has the highest sorption capacity. This correlates with BET analysis data, which showed specific surface areas of 62.39, 66.51, 74.19, 76.49, and 10.5 $m^2/cm^3$ for samples S1, S2, S3, S4, and S5, respectively.

**Table 4.** Kinetics parameters of Rhodamine B dye sorption onto the obtained samples ($Q_{exp}$ is the amount of adsorbed dyes obtained from experimental data).

| Parameter | S1 | S2 | S3 | S4 | S5 |
|---|---|---|---|---|---|
| $Q_{exp}$, mg/g | 7.35 | 6.15 | 4.84 | 3.67 | 2.45 |
| Pseudo-first-order | | | | | |
| $Q_e$, mg/g | 6.79 | 6.46 | 4.47 | 4.19 | 2.42 |
| $K_1$, | 0.13 | 0.05 | 0.09 | 0.18 | 0.07 |
| $R^2$ | 0.97 | 0.79 | 0.97 | 0.97 | 0.95 |
| Pseudo-second-order | | | | | |
| $Q_e$, mg/g | 9.11 | 8.08 | 43.38 | 14.07 | 6.64 |
| $K_2$, | 0.03 | 0.06 | 0.03 | 0.04 | 0.04 |
| $R^2$ | 0.62 | 0.98 | 0.36 | 0.3 | 0.26 |
| Intraparticle diffusion | | | | | |
| $K_d$, | 1.37 | 1.12 | 0.91 | 0.79 | 0.47 |
| C | 0.34 | −0.96 | 0.09 | −0.16 | −0.22 |
| $R^2$ | 0.96 | 0.82 | 0.98 | 0.86 | 0.96 |

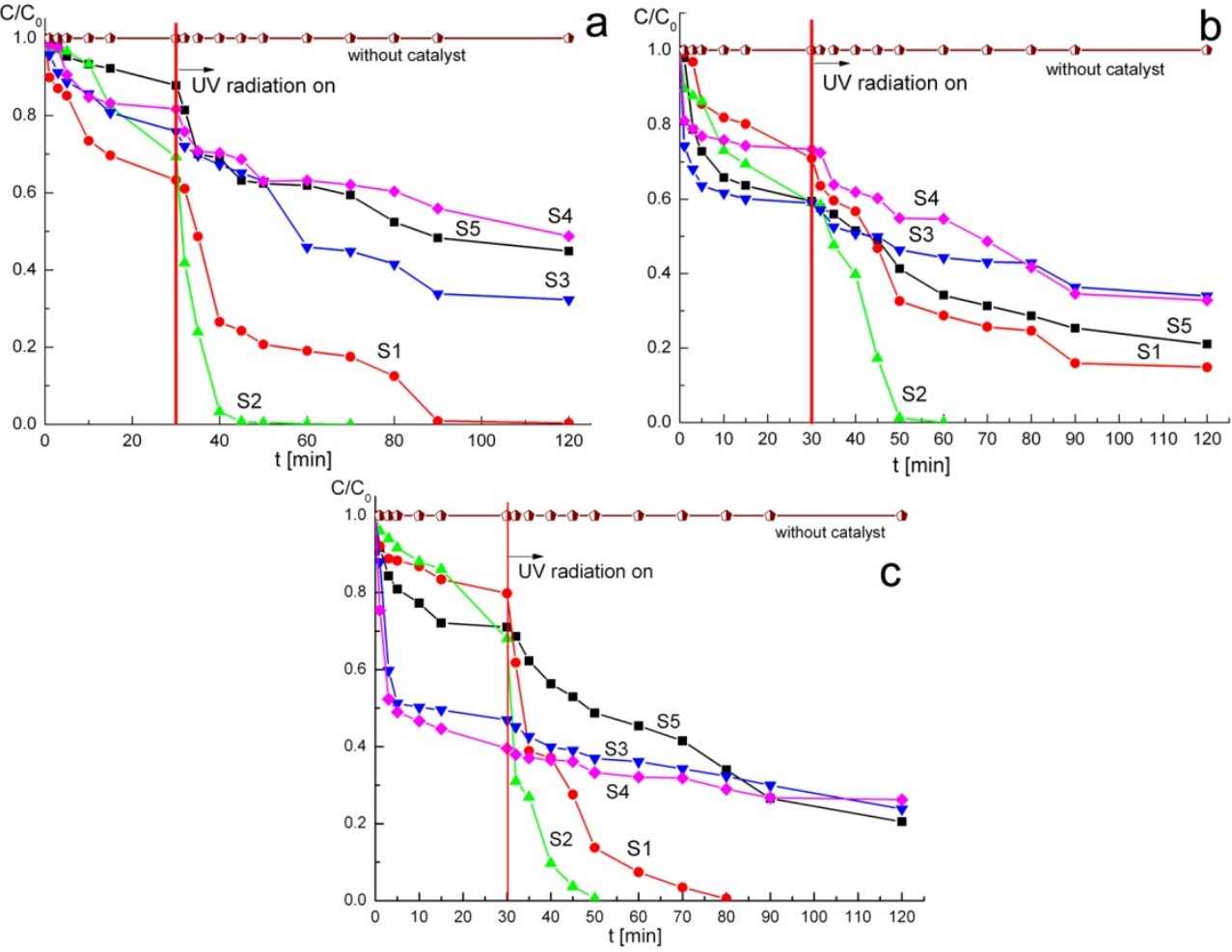

**Figure 6.** Photocatalytic performance of synthesized catalysts for destruction of Rhodamine B (**a**), Reactive Red 6C (**b**), and Methylene Blue (**c**).

**Table 5.** Kinetics parameters of Reactive Red 6C dye sorption onto the synthesized samples.

| Parameter | S1 | S2 | S3 | S4 | S5 |
|---|---|---|---|---|---|
| $Q_{exp}$, mg/g | 5.79 | 8.19 | 8.21 | 5.33 | 8.13 |
| Pseudo-first-order | | | | | |
| $Q_e$, mg/g | 5.72 | 7.56 | 4.35 | 2.56 | 7.23 |
| $K_1$, | 0.08 | 0.08 | 0.21 | 0.18 | 0.16 |
| $R^2$ | 0.88 | 0.96 | 0.88 | 0.82 | 0.94 |
| Pseudo-second-order | | | | | |
| $Q_e$, mg/g | 7.13 | 83.75 | 55.2 | 46.08 | 23.96 |
| $K_2$, | 0.04 | 0.08 | 0.01 | 0.01 | 0.04 |
| $R^2$ | 0.1 | 0.68 | 0.38 | 0.61 | 0.27 |
| Intraparticle diffusion | | | | | |
| $K_d$, | 1.16 | 1.51 | 1.28 | 0.79 | 1.65 |
| C | −0.48 | 0.09 | 2.89 | 2.02 | 0.51 |
| $R^2$ | 0.91 | 0.97 | 0.6 | 0.55 | 0.83 |

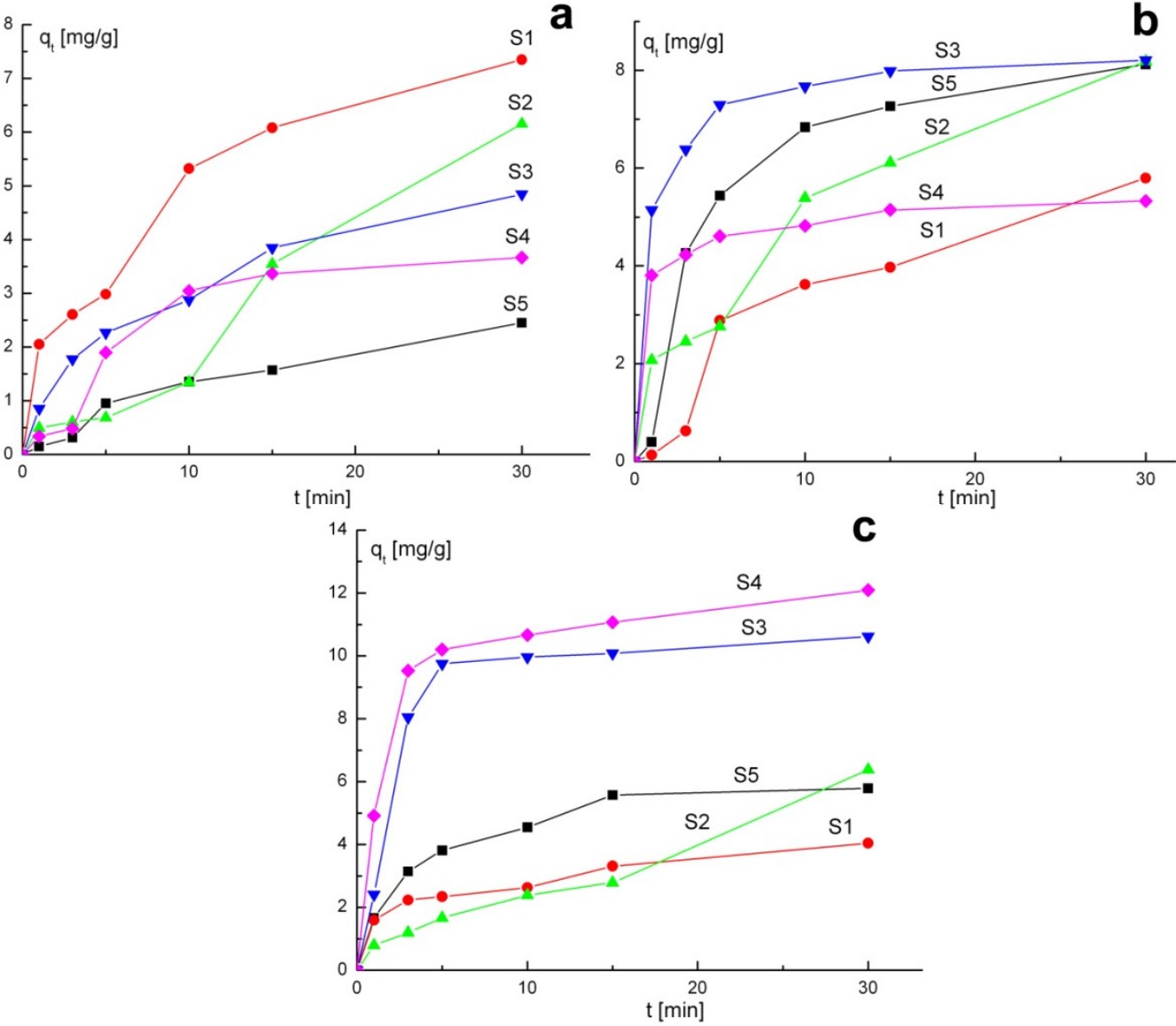

**Figure 7.** Kinetic curves of RhB (**a**), RR6C (**b**), and MB (**c**) absorption on synthesized samples at room temperature.

In experiments with Rhodamine B, the closeness of the values of $Q_{exp}$ (experimental sorption capacity) and $Q_e$ obtained by the pseudo-first-order approximation may mean that the sorption kinetics are described precisely by this model. However, for S2, the values of the $R^2$ coefficients are closer to 1 for the pseudo-second-order model. The same trend with correlation coefficients can be traced for the S1, S2, and S5 samples with the RR6C and MB dyes (see Tables 5 and 6). This may mean that sorption processes are described by models of different orders at different stages. An analysis of the sorption data showed that for S3 and S4, the sorption kinetics of RR6C and MB dyes is poorly described by standard models. However, these two dyes exhibit multi-linearity in the $q_t$ vs. $t^{1/2}$ coordinates (Figure 8). This shows that the adsorption processes of these dyes are affected by more than one process. The first step is the stage of external surface sorption or instantaneous diffusion, during which most of the dyes are sorbed on the external surface. The second stage is the slow phase. It is associated with the intraparticle diffusion of the dye molecules in the pores of the sorbent. Moreover, for both samples and two dyes, the lines pass through the center of the coordinates, which indicates that the sorption process can be controlled by intraparticle diffusion.

**Table 6.** Kinetics parameters of Methylene Blue dye sorption onto obtained samples.

| Parameter | S1 | S2 | S3 | S4 | S5 |
|---|---|---|---|---|---|
| $Q_{exp}$, mg/g | 4.04 | 6.38 | 10.61 | 12.09 | 5.79 |
| Pseudo-first-order | | | | | |
| $Q_e$, mg/g | 2.99 | 5.91 | 6.24 | 6.69 | 5.49 |
| $K_1$, | 0.09 | 0.04 | 0.19 | 0.15 | 0.19 |
| $R^2$ | 0.85 | 0.94 | 0.72 | 0.71 | 0.94 |
| Pseudo-second-order | | | | | |
| $Q_e$, mg/g | 94.97 | 32.69 | 171.23 | 41.32 | 9.99 |
| $K_2$, | 0.02 | 0.03 | 0.03 | 0.01 | 0.25 |
| $R^2$ | 0.66 | 0.62 | 0.1 | 0.16 | 0.31 |
| Intraparticle diffusion | | | | | |
| $K_d$, | 0.67 | 0.72 | 1.96 | 2.02 | 1.41 |
| C | 0.63 | 0.02 | 2.38 | 3.32 | 0.29 |
| $R^2$ | 0.90 | 0.99 | 0.65 | 0.68 | 0.97 |

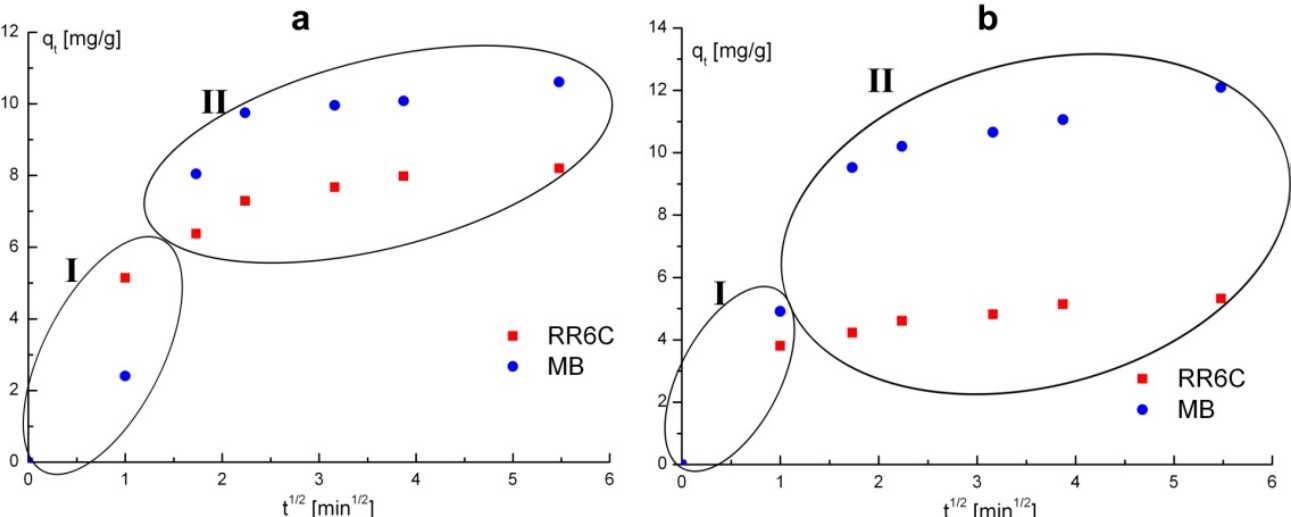

**Figure 8.** Plots of absorption capacity as function of $t^{1/2}$ for S3 (**a**) and S4 (**b**).

*3.4. Photocatalytic Activity*

The experimental results of photocatalysis kinetics showed that the sorption process can be predominant in the case of methylene blue dye on samples S3 and S4, as well as in the case of reactive red dye on samples S2, S3, and S5 (Figure 6). In other cases, the sorption process is not limiting. The removal efficiency of all three dyes on sample S2 is 100%. The time to achieve complete removal varies from 15 to 30 min of light exposure, depending on the type of dye. When using sample S1, it is also possible to achieve 100% removal efficiency of the MB dye and 99.7% of the RB dye but for a longer exposure time (50 and 60 min, respectively). Despite the large contribution of the sorption process for samples S3 and S4, for methylene blue, the degrees of photodecomposition are low and amount to 26 and 13%, respectively. The high photocatalytic activity of S2 can be explained by the phase composition, namely, the presence of only one copper oxide.

According to the presented data, only samples S1 and S2 can be considered as potential photocatalysts for the different types and classes of dyes. Samples S3 and S4 can be considered as promising sorbents.

### 3.5. Kinetic Studies of the Photocatalytic Destruction of Dyes

The kinetic curves of the photocatalytic decomposition of dyes are shown in Figure 9. Dependences $\ln(C/C_0) = f(t)$ were described by the pseudo-first-order of the reaction under the conditions of simplifying the description of the kinetics of heterogeneous processes [33]. As can be seen, the rate constants of decomposition reactions are different depending on the synthesis conditions and the nature of the dye (Table 7). The previously published data on studies of photocatalytic decomposition of methylene blue and Rhodamine B dyes using Cu–Fe binary oxide systems obtained by various methods are presented in Table 8. The comparison of the kinetic data of the decomposition of other dyes when using catalysts with a different composition (for example, based on $TiO_2$) will be incorrect since the processes of the destruction of the different classes of dyes proceed differently with the participation of different catalysts. According to the presented data, the photocatalysts S1 and S2 show excellent photocatalytic activity.

**Table 7.** Kinetic parameters of photocatalytic destruction of dyes.

| Parameter | S1 | S2 | S3 | S4 | S5 |
|---|---|---|---|---|---|
| | Rhodamine B | | | | |
| K, min$^{-1}$ | 0.0069 | 0.0637 | 0.1564 | 0.0108 | 0.0049 |
| R$^2$ | 0.90 | 0.98 | 0.93 | 0.92 | 0.91 |
| | Reactive Red 6C | | | | |
| K, min$^{-1}$ | 0.0187 | 0.1165 | 0.0059 | 0.0095 | 0.0125 |
| R$^2$ | 0.90 | 0.90 | 0.92 | 0.94 | 0.95 |
| | Methylene Blue | | | | |
| K, min$^{-1}$ | 0.0878 | 0.1129 | 0.0068 | 0.0048 | 0.0138 |
| R$^2$ | 0.98 | 0.90 | 0.97 | 0.93 | 0.98 |

**Table 8.** Summary of some previously published data in the photocatalytic activity of Cu–Fe binary oxides.

| Photocatalyst | Synthesis Method | Degradation Efficiency/Time | K, min$^{-1}$ | Ref. |
|---|---|---|---|---|
| | | Methylene Blue | | |
| Cu–Fe oxides | One-step sparking | 75% | | [34] |
| $\alpha$-Fe$_2$O$_3$/Cu$_2$O heterostructures | Hydrothermal precipitation | 55.5–81%/45 min | 0.025 | [35] |
| CuFe$_2$O$_4$ | Hydrothermal | 80–96%/60 min | | [36] |
| Fe(ox)-FeO/CuO | Chemical precipitation | 95%/100 min | | [37] |
| S1 | Underwater plasma | 100%/50 min | 0.0878 | This work |
| S2 | | 100%/20 min | 0.1129 | |
| | | Rhodamine B | | |
| Fe$_2$O$_3$/Cu$_2$O | Hydrothermal + sonication | 90%/120 min | 0.726 | [38] |
| CuFe$_2$O$_4$ | Sol-gel auto-combustion | 5%/30 min | 0.0006 | [39] |
| CuFe$_2$O$_4$ | Hydrothermal | 50%/140 min | | [40] |
| CPF/CuFe$_2$O$_4$ | Coprecipitation | 98.2%/80 min | 0.0235 | [41] |
| S1 | Underwater plasma | 99.7%/60 min | 0.0069 | This work |
| S2 | | 100%/15 min | 0.0637 | |

We consider the mechanism of photocatalysis for sample S2 showing excellent photo-catalytic activity towards all dyes. According to the data in Figure 3, the phase composition of the sample is a mixture. The photocatalytic mechanism is based on the generation of an electron-hole pair. The $E_{VB}$ of $Cu_2O_3$, $\alpha$-$Fe_2O_3$, and FeO are +1.87, +2.88, and +2.13 eV/NHE. Additionally, the $E_{CB}$ of $Cu_2O_3$, $\alpha$-$Fe_2O_3$, and FeO are +0.31, +0.08, and −0.09 eV/NHE which can be calculated by a formula presented elsewhere [42]. The probable mechanism starts with photo-generated $e^-/h^+$ pairs from $Cu_2O_3$ (Figure 10). Photo-generated electrons in $Fe_2O_3$ and FeO are injected into the CB of $Cu_2O_3$. At the same time, photo-excited holes will move towards the surface of the iron. It can improve charge separation and inhibits electron-hole recombination. It is a key factor in high photo-catalytic activity. For other samples, $Cu_2O$, CuO, and $Fe_3O_4$ will be added to the energy level diagram.

After the photocatalysis procedure, the catalyst was easily removed from the solution using a magnet. The solid phase was dried at 200 °C for 1 h. The weight loss was 3–5%. A series of experiments were carried out on the reuse of catalysts in the photocatalysis process (including the dark phase). The efficiency of the photodegradation of the dye Rhodamine B at three cycles of use for 1.5 h is shown in Figure 11. It was noted that the time for full decomposition of the dye increases by 1.5–2.5 times for S1 and S2 samples. The efficiency of photodecomposition decreases by 15–20% due to a decrease in the sorption capacity.

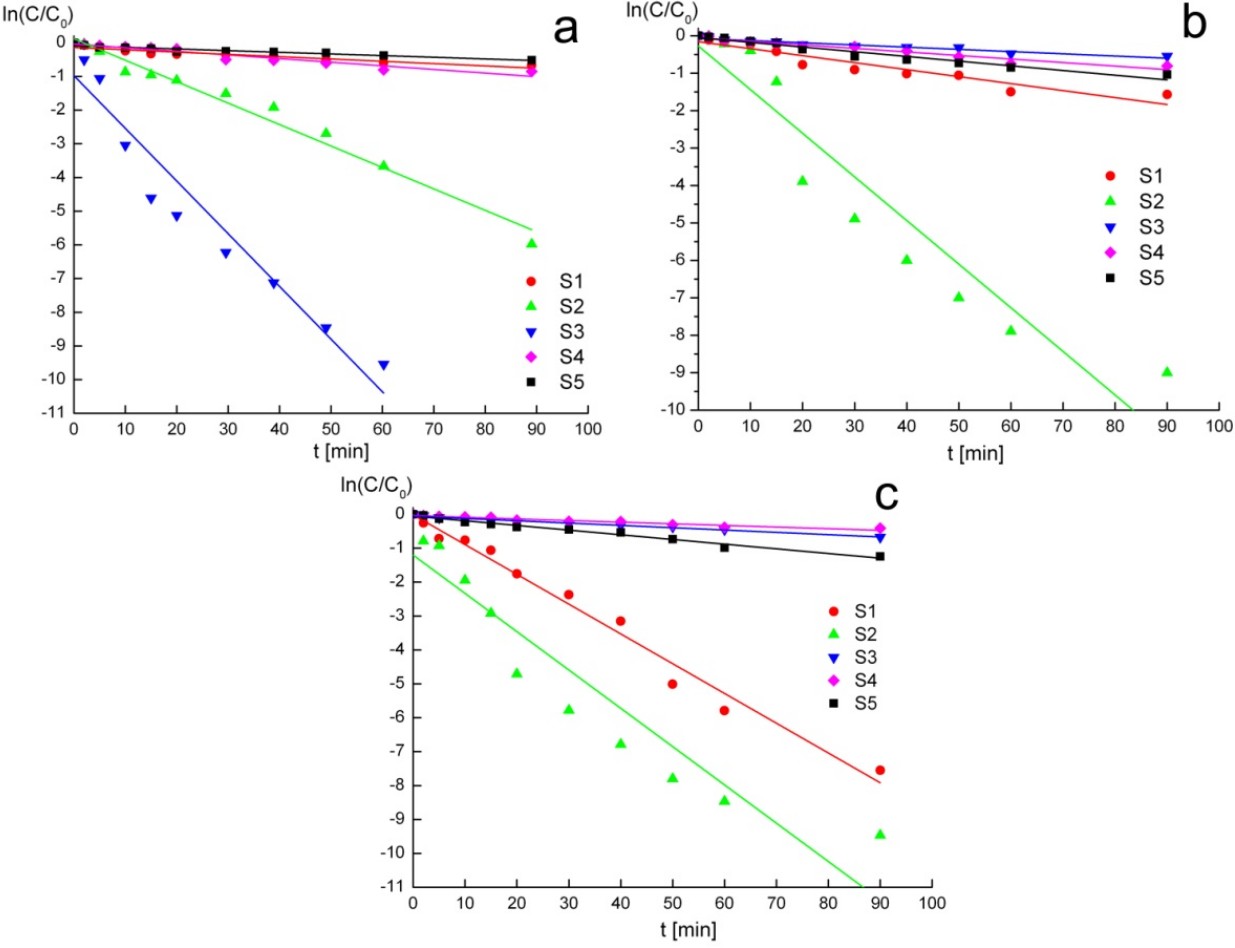

**Figure 9.** The kinetic curves of the photocatalytic decomposition of Rhodamine B (**a**), Reactive Red 6C (**b**), and Methylene Blue (**c**).

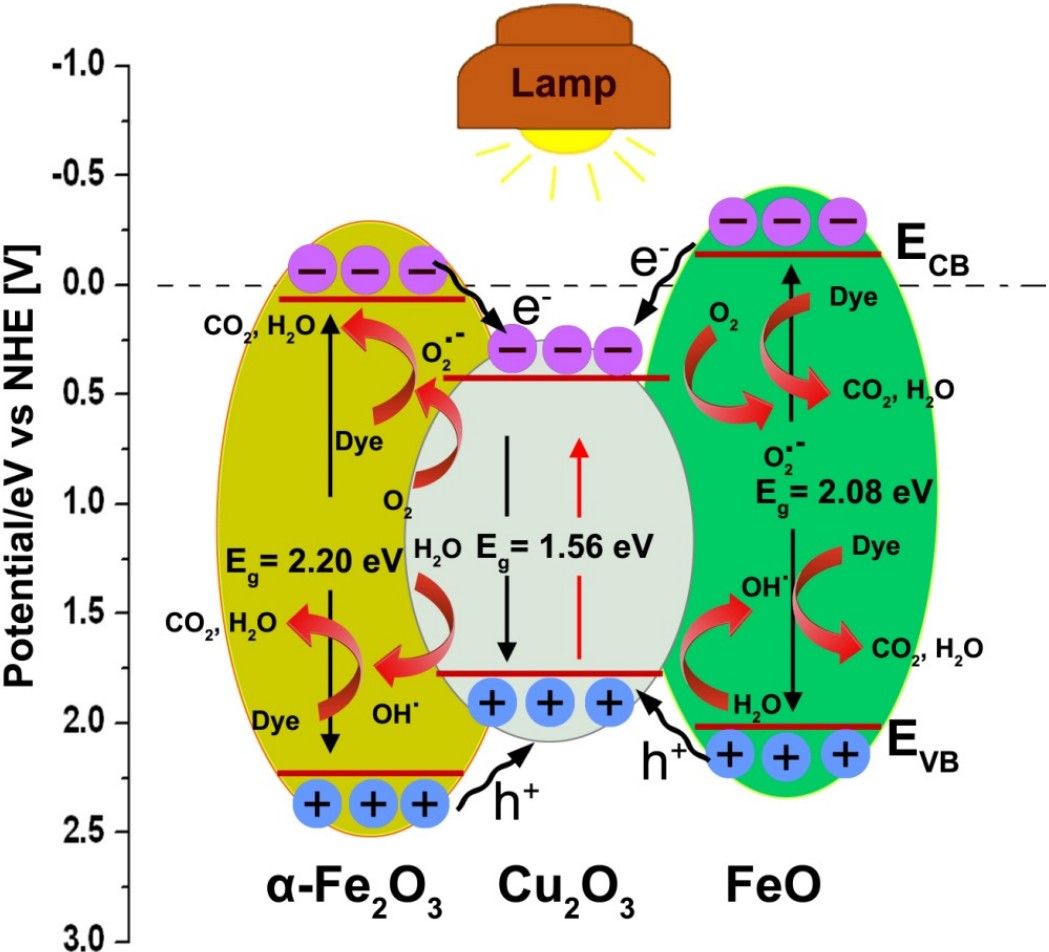

**Figure 10.** Schematic diagram for photocatalytic mechanism for S2.

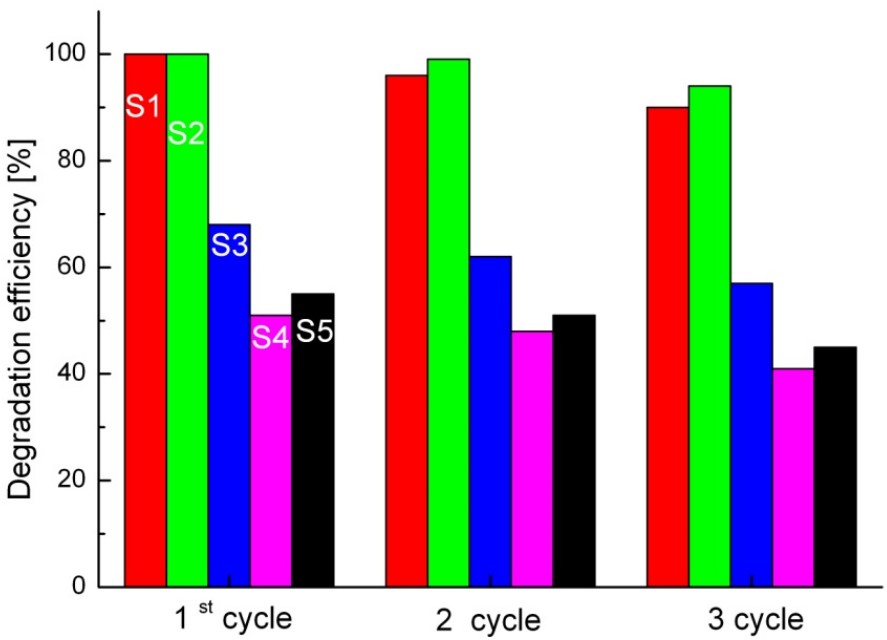

**Figure 11.** Reusability of prepared photocatalysts.

## 4. Conclusions

A comparison of two methods for the synthesis of chemical combustion and underwater plasma showed that both routes do not allow for obtaining Cu–Fe binary oxides. Due to the mobility of the plasma zone and the use of one type of fuel during combustion, systems of the mixtures of copper and iron oxides with different degrees of oxidation were obtained. The obtained materials showed high sorption and photocatalytic activity for different classes of dyes. The reuse of structures showed that the decrease in efficiency occurs due to a decrease in sorption capacity.

**Author Contributions:** Conceptualization, A.A.; methodology, A.K.; investigation, N.S., A.E. and A.L.; writing—original draft preparation, A.K. and N.S.; writing—review and editing, A.K. and N.S.; supervision, A.A.; project administration, A.A. and V.T. All authors have read and agreed to the published version of the manuscript.

**Funding:** This research was funded by the frame of the Government Assignment of the Ministry of Education and Science of Russia (no. 0092-2019-0003).

**Data Availability Statement:** Not applicable.

**Acknowledgments:** This study was performed in the frame of the Government Assignment of the Ministry of Education and Science of Russia (no. 0092-2019-0003). The authors would like to thank N. Fomina for performing the XRD analysis and SEM analysis, S. Guseinov for conducting the DSC analysis, N. Kochkina for conducting the DLS measurements, and A. Bazanov for performing the BET analysis at the center of the joint use of scientific equipment (the Upper Volga Regional Center for Physical-Chemical Research, Russia).

**Conflicts of Interest:** The authors declare no conflict of interest.

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
