# Peer review of "Sorption and Photocatalytic Characteristics of Composites Based on Cu–Fe Oxides"

_2673-7167, doi:10.3390/physchem2040022_

Round 1

Reviewer 1 Report

The paper from Anna Khlyustova and coworkers details the synthesis Cu-Fe oxides-based nanomaterials. In particular, the sorption and photocatalysis properties of these nanomaterial are studied and demonstrated. The author identify that the nanomaterials are able to degrade the dyes in solution with radiation. However, the concluded innovations are not clear enough. Several statements in manuscript need further clarification. 

1. Please check the equation 4 (page 3, line 110), if the qe should be qe2 in the term of k2qe

2. In the result of evidence of electrode’s sputtering, the spectrum contains more lines of atomic iron when Fe is in anode. I would like to question why more lines can be observed or if the author can provide citations?

3. In 158 lines (page 5), the author mentions Cu2O3+α-Fe2O3+Fe3O4 were formed. However, the peak in the spectrum was labeled as FeO. Can author provide the reason for the difference?

4. The size of particles from SEM is far smaller than the sizes from DLS. I would like to question why the difference can be observed. 

5. In the result of kinetics of dye’s absorption, the author just analysis the sorption data from S4 and S5. Can author provide reason for skipping S1, S2 and S3? 

6. In line 212 (page 9), the author uses a new label Qexp. What is the meaning of the new label?

7. The paper does not have conclusion part. I believe it is necessary, otherwise, it is difficult for readers to get the conclusion of the research efficiently. 

Author Response

  1. Please check the equation 4 (page 3, line 110), if the qe should be qe2 in the term of k2qe

Thank you for your comment. When writing expressions for the kinetics of the pseudo-first and pseudo-second order, qe2 is not denoted in the term k2qe since it is customary to bring all numerical values in tables with a clear distinction between qe to the pseudo-first or pseudo-second order.

  1. In the result of evidence of electrode’s sputtering, the spectrum contains more lines of atomic iron when Fe is in anode. I would like to question why more lines can be observed or if the author can provide citations?

Thanks to your comment. When the anode is made of iron wire a large number of emission lines of iron atoms are observed in the emission spectra of discharges. This may be due to the fact that iron is sprayed more with this electrode configuration (Table 1). In addition, the iron atoms sputtered from the anode can receive additional thermal excitation due to the higher anode temperature. Some lines are not recorded in the discharge radiation spectrum due to low intensity due to the low concentration of iron atoms in the plasma due to less cathode sputtering when iron is used as a cathode.

  1. In 158 lines (page 5), the author mentions Cu2O3+α-Fe2O3+Fe3O4 were formed. However, the peak in the spectrum was labeled as FeO. Can author provide the reason for the difference?

Thank you for your comment. The authors apologize for the typo. Yes, of course, FeO is present in the sample. Changes have been made to the text of the manuscript (highlighted in yellow).

  1. The size of particles from SEM is far smaller than the sizes from DLS. I would like to question why the difference can be observed.

Thank you for your comment. The dynamic light scattering method shows the hydrodynamic particle size. In this case, the particle size also includes the hydration shell around it. Thus, the particle size values obtained from the DLS method can be significantly larger than the particle sizes obtained from the results of SEM.

  1. In the result of kinetics of dye’s absorption, the author just analysis the sorption data from S4 and S5. Can author provide reason for skipping S1, S2 and S3?

The authors are grateful for the critical remark. We apologize for the incorrect information presented in tables 5 and 6. Changes have been made to the text of the manuscript regarding the description of the dye adsorption kinetics for all samples.

  1. In line 212 (page 9), the author uses a new label Qexp. What is the meaning of the new label?

Thanks for your comment. The explanation of the new label Qexp was added to the text (in yellow).

  1. The paper does not have conclusion part. I believe it is necessary, otherwise, it is difficult for readers to get the conclusion of the research efficiently. 

Thank you for your valuable comment. We have inserted the final part in the text (in yellow).

Reviewer 2 Report

In this work, the authors compare the characteristics of materials obtained by chemical combustion and under-water plasma and also study their sorption and photocatalytic properties.

This is an interesting work; Nevertheless some minor revisions are needed in order to publish this manuscript.

1) What is the light intensity of the 250 W mercury lamp was used as the source of light irradiation in photocatalysis experiments?

2) Can the authors comment on the re-use of their samples? Could they reuse them in order to check this parameter?

3) Could the authors compare the Kinetics parameters with analogous values in the literature and comment?

Author Response

1) What is the light intensity of the 250 W mercury lamp was used as the source of light irradiation in photocatalysis experiments?

Thanks for comment. The light intensity of this mercury lamp in the spectral region of 280-400 nm is 75 W/m2

2) Can the authors comment on the re-use of their samples? Could they reuse them in order to check this parameter?

Thanks for the helpful note. The authors carried out a series of experiments on the reuse of synthesized samples as photocatalysts. The results are added to the text of the manuscript (highlighted in yellow).

3) Could the authors compare the Kinetics parameters with analogous values in the literature and comment?

Thanks for the comment. In the previous version of the article, we did not provide comparative data on the kinetics since there are very few published data on the studies of the sorption properties of the Cu-Fe oxide systems. We supplemented our results on photodecomposition kinetics and compared them with published data.

Reviewer 3 Report

The presented Manuscript entitled “Sorption and photocatalytic characteristics of composites based 2 on Cu-Fe oxides” describes the comparison of the materials characteristics obtained by chemical combustion and under water plasma. In addition studied their sorption and photocatalytic properties.

1-     The introduction needs to be improved. “This simultaneously allows both removing the pollutants and decomposing it into harmful compounds.” it's not always is decomposing em harmful compounds. (see:  Catalysts 2019, 9, 343; doi:10.3390/catal9040343 )

2-      Many authors indicate the efficiency of TIO2 without doping, so the sentence must be rewritten. .

3-     The author could cite other methods of preparing TiO2, which are efficient.

4-     The background of the research group, showing previous published results and highlighting the novelty of this work compared to previous own papers and other publications.

Materials and Methods

Methods of synthesis

Are the conditions for synthesis of the material new? If it is not necessary to insert references.

2.3 Photocatalytic activity – Authors must justify the use of dyes for photocatalysis tests. The use of dyes in photocatalysis is very problematic. (see: DOI: 10.1021/acs.est.6b00213).

How to be sure that what is acting on the degradation is the catalyst + radiation and not just the radiation? Authors should add photolysis tests.

 The manuscript is well,  but some questions need to be clarified, so I consider the work interesting for publication, after major revisions

Author Response

  • The introduction needs to be improved. “This simultaneously allows both removing the pollutants and decomposing it into harmful compounds.” it's not always is decomposing em harmful compounds. (see:  Catalysts 2019, 9, 343; doi:10.3390/catal9040343 )

Thanks for the critical comment. The authors revised the text of the Introduction and made amendments according to the remark. 

  • Many authors indicate the efficiency of TIO2 without doping, so the sentence must be rewritten.

The authors rewrote the sentence according to the remark. 

  • The author could cite other methods of preparing TiO2, which are efficient.

Thanks for the note. But titanium dioxide is not considered in this work. However, we have added to the text a mention of different methods for preparing Cu-Fe oxides. 

  • The background of the research group, showing previous published results and highlighting the novelty of this work compared to previous own papers and other publications.

Thanks for the comment. The authors added the novelty of the presented work compared to previously published ones. 

Materials and Methods

Methods of synthesis

1. Are the conditions for synthesis of the material new? If it is not necessary to insert references.

The conditions for fusion using underwater plasma are not new. It has been described and presented in more detail in reference (23). This has been added to the text of the manuscript.

2.3 Photocatalytic activity – Authors must justify the use of dyes for photocatalysis tests. The use of dyes in photocatalysis is very problematic. (see: DOI: 10.1021/acs.est.6b00213).

Thanks for the critical comment. The authors got acquainted with the article proposed by the reviewer. The choice of dyes for photocatalysis was determined by the fact that we chose representatives of different classes of dyes. Rhodamine B is a fluorone dye that exists in solution as a zwitterion. Active Red 6C is a monoazo dye, which is an anion in solution. Methylene blue is a thiazine dye. It presents in solution as a cation. The toxicity of an individual dye is difficult to assess, since these dyes are not used in the food industry. But in the textile industry, different classes of dyes are used, and the wastewater from such enterprises contains a mixture of various dyes. The study of sorption and photocatalysis of different materials is the subject of further research. The authors inserted an appropriate explanation of the choice of dyes for photocatalysis in the text of the manuscript.

3. How to be sure that what is acting on the degradation is the catalyst + radiation and not just the radiation? Authors should add photolysis tests.

Thanks for helpful comment. Photolysis experiments (without adding a catalyst) were performed, the results are shown in Figure 6.

Reviewer 4 Report

1. How did the authors optimize the catalyst loading, initial conc. and pH ?

2. Photocatalytic mechanism needs to be discussed in detail wrt to FeOx/CuOx.

3. The photocatalytic results needs to be compared with Cu-Fe oxides prepared by other synthetic routes.

Author Response

  1. How did the authors optimize the catalyst loading, initial conc. and pH ?

Thanks for the comment. Based on published data, it is known that the larger the catalyst loading, the faster and more efficiently the photocatalysis process proceeds. We have studied the effect of low loading doses (cost minimization). The initial concentration of dyes was chosen as the starting point (minimum value).  Moreover, the dyes methylene blue and Rhodamine B have the tendency to aggregate into dimer structures at certain concentrations (Fornili, S. L., Sgroi, G., & Izzo, V. (1981). Solvent isotope effect in the monomer–dimer equilibrium of methylene blue. Journal of the Chemical Society, Faraday Transactions 1: Physical Chemistry in Condensed Phases, 77(12), 3049-3053; Arbeloa, I. L., & Ojeda, P. R. (1982). Dimeric states of rhodamine B. Chemical Physics Letters, 87(6), 556-560). In further work, it is planned to increase the concentrations to the values of the content of dyes in real wastewater. It is known that most dyes react to changes in the acidity of the solution, acting as an indicator. Therefore our experiments were carried out at neutral pH since real wastewater have pH of 6.5-8.

  1. Photocatalytic mechanism needs to be discussed in detail wrt to FeOx/CuOx.

Thanks for the helpful note. The mechanism of photocatalysis is present in almost all articles devoted to photocatalytic research. We also included a discussion of the mechanism taking into account our data on the phase composition of the catalysts.

  1. The photocatalytic results needs to be compared with Cu-Fe oxides prepared by other synthetic routes.

Thanks for helpful comment. We have added a comparison of photocatalysis kinetics data between our results and published data.

Round 2

Reviewer 1 Report

The manuscript has been well revised and I now support publication.

Reviewer 3 Report

The authors have reviewed the manuscript, in my opinion it may be accepted for publication.

Reviewer 4 Report

Authors have improved the manuscript by including the necessary details that I asked for. The overall comparison has been included with other previously published work Cu-Fe oxides and the mechanism w.r.t to potential bands. Recommend acceptance in its present form.